# Umbilical Cord PRP vs. Autologous PRP for the Treatment of Hip Osteoarthritis

**DOI:** 10.3390/jcm11154505

**Published:** 2022-08-02

**Authors:** Alessandro Mazzotta, Enrico Pennello, Cesare Stagni, Nicolandrea Del Piccolo, Angelo Boffa, Annarita Cenacchi, Marina Buzzi, Giuseppe Filardo, Dante Dallari

**Affiliations:** 1Reconstructive Orthopaedic Surgery Innovative Techniques—Musculoskeletal Tissue Bank, IRCCS Istituto Ortopedico Rizzoli, 40136 Bologna, Italy; alessandro.mazzotta@ior.it (A.M.); enrico.pennello@ior.it (E.P.); cesare.stagni@ior.it (C.S.); nicolandrea.delpiccolo@ior.it (N.D.P.); dante.dallari@ior.it (D.D.); 2Applied and Translational Research (ATR) Center, IRCCS Istituto Ortopedico Rizzoli, 40136 Bologna, Italy; ortho@gfilardo.com; 3Single Metropolitan Transfusion Service, AUSL Bo, 40136 Bologna, Italy; annarita.cenacchi@ior.it; 4Emilia Romagna Cord Blood Bank, Department of Pathology, IRCCS Azienda Ospedaliero Universitaria di Bologna, 40138 Bologna, Italy; marina.buzzi@aosp.bo.it

**Keywords:** hip, injection, osteoarthritis, platelet-rich plasma, PRP, umbilical cord

## Abstract

Umbilical cord platelet-rich plasma (C-PRP) has more growth factors and anti-inflammatory molecules compared with autologous PRP (A-PRP) derived from peripheral blood. The aim of this study was to compare intra-articular C-PRP or A-PRP injections in terms of safety and clinical efficacy for the treatment of patients with hip osteoarthritis (OA). This study investigated the results of 100 patients with hip OA treated with three weekly ultrasound-guided injections of either C-PRP or A-PRP. Clinical evaluations were performed before the treatment and after two, six, and twelve months with the HHS, WOMAC, and VAS scores. No major adverse events were recorded. Overall, the improvement was limited with both treatments. Significant improvements in VAS (*p* = 0.031) and HHS (*p* = 0.011) were documented at two months for C-PRP. Patients with a low OA grade (Tonnis 1-2) showed a significantly higher HHS improvement with C-PRP than A-PRP at twelve months (*p* = 0.049). C-PRP injections are safe but offered only a short-term clinical improvement. The comparative analysis did not demonstrate benefits compared with A-PRP in the overall population, but the results are influenced by OA severity, with C-PRP showing more benefits when advanced OA cases were excluded. Further studies are needed to confirm the most suitable indications and potential of this biological injective approach.

## 1. Introduction

Hip osteoarthritis (OA) is a degenerative joint disease with a chronic course, characterized by regressive changes in the articular hyaline cartilage and modifications of the bony (sub-chondral bone), synovial, and capsulo-ligamentous components [1]. Hip OA is clinically characterized by pain and progressive functional limitation, and over time it can affect patient life habits up to disability [1]. Symptomatic hip OA affects 9.3% of women and 8.7% of men over the age of 45, with a significant social and economic impact, both in terms of reduction or suspension of work and in terms of health costs [2,3]. Conservative strategies include nonsteroidal anti-inflammatory drug (NSAID) treatment, physical therapy, and intra-articular corticosteroids or hyaluronic acid injections. These options can offer modest benefits with efficacy decreasing over time, leading in the end to the need for total hip arthroplasty (THA) [4]. Although THA was valued as the operation of the century, considering satisfactory results obtained and durability over the years [5,6] it is associated with complications such as dislocation, infection, and the need for reoperation [7,8]. This explains the research efforts to find new solutions to address hip OA, looking for products with disease-modifying effects capable of delaying or avoiding surgery.

Platelet-rich plasma (PRP) gained significant interest in the last decade for the treatment of joints with cartilage degeneration and OA [9], thanks to its safety and low costs, and the simple preparation technique to obtain its biologically active content [10]. PRP’s potential depends on the large number of growth factors, cytokines, and bioactive molecules contained in the platelet alpha-granules [11]. Through platelet activation and their release, these molecules act at the level of cartilage tissue and synovium, attenuating cartilage damage progression and reducing synovial inflammation [12,13]. Autologous PRP (A-PRP) is the most common blood derivative used in clinical practice for the treatment of OA. However, it can also contain pro-inflammatory cytokines [14], and its quality may depend on many factors related to the patient such as the quantity and quality of platelets, the concomitant therapies taken [15], and the patient’s age, which is often advanced in patients affected by OA [16,17]. Thus, to overcome the limits of A-PRP, PRP obtained from the umbilical cord (C-PRP) has been recently investigated. Previous studies showed that C-PRP has more growth factors and anti-inflammatory molecules compared with PRP derived from peripheral blood [18]. Nevertheless, no clinical studies investigated the biological advantages of C-PRP over A-PRP in terms of clinical improvement in the treatment of hip OA.

The aim of this study was to compare intra-articular C-PRP or A-PRP injections in terms of safety and clinical efficacy for the treatment of patients with symptomatic hip OA.

## 2. Materials and Methods

### 2.1. Study Design and Selection Criteria

After approval of the study protocol by the local ethics committee (Prot. number 0033724), between October 2013 and October 2014, 174 consecutive patients with unilateral symptomatic hip OA were screened. The inclusion criteria for the recruitment were: age between 18 and 65 years; unilateral hip pain and functional impairment from at least four months, with pain intensity at baseline of at least 20 on a 100 mm visual analogue scale (VAS); body mass index (BMI) < 35; failure of conservative treatment. Each patient was radiographically evaluated within one month before the treatment and only patients with low or intermediate OA (Tonnis 1-3) were included. Exclusion criteria were: patients under the age of 18 or over 65; patients unable to provide informed consent to the experimental protocol; patients with systemic disorders (bleeding disorders, cardiovascular disease, infections, and immune system disorders); patients suffering from neoplastic or local infectious problems; patients with OA secondary to protrusio acetabuli or an excessive deformity (hip dysplasia, collapse deformity, osteonecrosis, OA secondary to the sequelae of Perthes or epiphysiolysis); patients with OA secondary to rheumatoid arthritis; patients with local skin lesions; and pregnant women. Patients who did not meet the necessary requirements for blood harvesting to produce PRP (hemoglobin level < 11 g/dL or platelet count < 150,000/mm^3^ at blood harvest) or patients who refused to undergo transfusions of homologous blood components due to problems of a religious or ideological nature were also excluded from the study. All patients signed a detailed informed consent form. Patients undergoing injective therapy with C-PRP treatment gave signed consent to homologous transfusion therapy. All patients underwent blood-type definition and compatibility (ABO-Rh).

From the institutional database, an independent statistician performed a match-paired selection, blinded to the study outcome, of patients treated with A-PRP or C-PRP: 50 patients were treated with 3 weekly ultrasound-guided injections of A-PRP and 50 patients were treated with 3 weekly ultrasound-guided injections of C-PRP. Four patients in the C-PRP group did not complete the follow-up and were excluded from the final analysis. Thus, 50 patients in the A-PRP group and 46 patients treated with C-PRP were included (Figure 1). The statistical analysis did not reveal any demographic differences at baseline between the two groups, as reported in Table 1. Likewise, the clinical-radiographic features of the included patients between the two groups were comparable, except for the Harris Hip Score (HHS) which showed a significantly higher score (*p* = 0.026) for the A-PRP group (83.8 ± 11.9) than the C-PRP group (78.6 ± 11.4).

All the patients were evaluated at baseline and prospectively at 2, 6, and 12 months of follow-up using the HHS, the Western Ontario and McMaster Universities Osteoarthritis Index (WOMAC), and the VAS pain score. At baseline, all patients underwent weight-bearing antero-posterior radiography of the pelvis and were classified for the OA grade according to the Tonnis classification.

### 2.2. A-PRP Production

A-PRP was produced via a manual method without the use of a commercial kit. The procedure began with the collection of a bag of 150 mL of autologous blood. The bag was centrifuged at 1800 rpm for 15 min. Through a closed circuit, plasma and buffy coat were transferred to a second bag; red blood cells were thus removed. The bag was centrifuged at 3500 rpm for 10 min; the supernatant was removed in order to produce 20 mL of PRP. The goal was to obtain a total number of platelets increased by 4–5 fold compared with the baseline and a mean platelet concentration of 1000 × 10^9^/L ± 20%. PRP was divided into three small aliquots of 5 mL each, and a sample for testing. The three units were stored at −30 °C. For each treatment, units were thawed in a dry thermostat at 37 °C for 30′ just before application (injection). After thawing, the PRP sample was transferred directly from the transfusion unit to the outpatient clinic in the same hospital, using a thermal bag and avoiding exposure to light. Before each injection, 10% of Ca-gluconate was added to PRP concentrate to activate platelets.

### 2.3. C-PRP Production

C-PRP was produced via a manual method without the use of a commercial kit. All steps from the recruitment to the processing and registration of Cord Blood (CB) were performed by Emilia Romagna Cord Blood Bank (ERCB) according to guidelines edited by the Foundation for the accreditation of cellular therapy (FACT) and the Italian regulation. CB collection was performed when the placenta was still in utero by puncturing the umbilical vein with a sterile system (Cord blood collection set, JMS, Singapore) in a bag containing 20 mL citrate-phosphate-dextrose (CPD). CB was collected from spontaneous term births free of complications and from Caesarean births (≥37th week of pregnancy), decided by trained and qualified health personnel. The units underwent a series of checks and tests to establish the blood characteristics and its suitability for preservation and therapeutic use. Maternal infectious disease markers (HIV, HCV, HBV Treponema pallidum, CMV, Toxoplasmosis, and HTLV-I/II) evaluations were performed.

The obtained CB units were processed within 48 h. The procedure involves the following phases: preparation of a CB pool consisting of 4–5 homogroup units having a final weight between 300 and 410 g: the individual bags are made to flow through a sterile connection into a single collection bag making up the kit of separation (Fresenius, Terumo); centrifugation of the CB pool in a Heraeus Cryofuge 6000i centrifuge at the rate of 799× *g*, for a time of 4 min in order to obtain the maximum platelet content in the PRP and the minimum residual platelet in the waste; separation of the PRP from concentrated red blood cells and leukocytes with Compomat G4 automatic separators from the Fresenius Kabi company during which the PRP is pushed into the second bag making up the kit, after in-line filtration that retains the leukocytes; concentration of the PRP obtained by further centrifugation of the same at 3845× *g* for a time of 6 min and subsequent transfer of excess plasma into a sterile connected bag by manual crushing of the original bag. The goal was to obtain a total number of platelets increased by 4–5 fold compared with the baseline and a mean platelet concentration of 1000 × 10^9^/L ± 20%. The PRP was divided in 5 mL aliquots into small bags and frozen at −80 °C. Microbiological tests were performed (Bact Alert Biomerieux) for each preparation. Similar to A-PRP, the activation of the C-PRP occurred at the time of use on the patient by adding 10% calcium gluconate.

### 2.4. Injective Technique

In both groups, a convex probe (2- to 5-MHz convex transducer MicroMaxx Ultrasound System; SonoSite Inc., Bothell, WA, USA) was used for the ultrasound-guided procedure. The patient was placed in a supine position. A preliminary ultrasound examination was performed to set focus, depth, and frequency. A sterile field was set up with 2% chlorhexidine. Sterile conduction gel and a sterile probe cover were used for the probe. Chlorhexidine itself or sterile gel was used as a transducer. The probe was positioned in line with the femoral neck, the skin was pricked with the needle placed at about 45° of inclination and proceeding deeply until reaching the head-neck passage of the proximal femur. Once the needle was positioned, the internal core was removed and the previously activated PRP portion was injected.

The patient was then discharged with the recommendation to avoid prolonged activities and overloads for 1 or 2 days and with the advice to apply local ice in the hours following the treatment. In case of pain, the use of paracetamol was allowed, but not that of NSAIDs.

### 2.5. Statistical Analysis

All continuous data were expressed in terms of the mean and the standard deviation of the mean; the categorical data were expressed as frequency and percentages. The Shapiro–Wilk test was performed to test the normality of continuous variables. The Levene test was used to assess the homoscedasticity of the data. The Repeated Measures General Linear Model (GLM) with Sidak test for multiple comparisons was performed to assess the differences at different follow-up times. The ANOVA test was performed to assess the between-groups differences in continuous, normally distributed, and homoscedastic data; the Mann–Whitney non-parametric test was used otherwise. The ANOVA test, followed by the post hoc Sidak test for pairwise comparisons, was performed to assess the differences among groups in continuous, normally distributed, and homoscedastic data; the Kruskal–Wallis non-parametric test, followed by the post hoc Mann–Whitney test with Bonferroni correction for multiple comparisons, was used otherwise. The Spearman rank correlation was used to assess correlations between numerical scores and continuous data; the Kendall Tau-b ordinal correlation was used to assess correlations between ordinal data. The Fisher Chi-square exact test was performed to assess the relationship between dichotomous variables. The Pearson Chi-square evaluated using an exact test was performed to investigate relationships between categorical variables. For all tests *p* < 0.05 was considered significant.

All statistical analysis was performed using SPSS v.19.0 (IBM Corp., Armonk, NY, USA).

## 3. Results

In both treatment groups, there were no major adverse events related to the injection procedures during the treatment and follow-up periods. Five patients in the A-PRP group (10.0%) and two patients in the C-PRP group (4.3%) had mild adverse events, reporting pain in the days following the injection procedure, which resolved within a few days after ice therapy, rest, and paracetamol. There was no statistically significant difference between the two groups in terms of adverse events. During the study period, three patients in the A-PRP group (6.0%) and eight patients in the C-PRP group (17.4%) required a new treatment for symptom persistence (in the A-PRP group one patient underwent total hip replacement, one patient underwent mini-open surgery, and one patient preferred to opt for injective treatment with hyaluronic acid; in the C-PRP group five patients underwent total hip replacement, one patient underwent arthroscopic surgery, and two patients preferred to opt for injective treatment with hyaluronic acid). No statistically significant difference in terms of treatment failures was observed.

Both groups reported an overall not statistically significant improvement from baseline to all follow-ups for all clinical outcomes (Figure 2, Figure 3 and Figure 4). In the C-PRP group, a statistically significant improvement was observed from baseline to 2 months of follow-up for the VAS pain (from 37.2 ± 22.0 to 28.5 ± 22.1, *p* = 0.011) and for the HHS (from 78.6 ± 11.4 to 83.3 ± 13.3, *p* = 0.031). This improvement was not confirmed at 6 and 12 months of follow-up for both scores. No statistically significant differences were observed between the groups in terms of absolute values and improvement of all clinical scores at all follow-up evaluations (Table 2).

Further analysis was performed to determine the parameters that influenced the clinical outcomes at follow-up. Sex, age, and previous surgery did not significantly influence the post-injective clinical outcome. Conversely, the OA severity based on the Tonnis classification influenced the clinical outcomes of both groups in terms of VAS pain at 12 months (tau = 0.201, *p* = 0.017) and WOMAC score at 12 months (tau = 0.190, *p* = 0.020).

Sub-analyses were then carried out considering patients with a low OA grade (Tonnis 1-2) compared with patients with a medium OA grade (Tonnis 3) (Figure 5). In the A-PRP group, a higher improvement was observed for patients with Tonnis 1-2 in terms of VAS pain (7.6 ± 14.0 vs. −3.6 ± 18.1, *p* = 0.041) and WOMAC score (6.8 ± 14.2 vs. −0.1 ± 10.7, *p* = 0.027) compared with patients with Tonnis 3 at 12 months. In the C-PRP group a similar trend was documented, although it did not reach statistical significance for patients with Tonnis 1-2 compared with patients with Tonnis 3 in terms of VAS pain (12.4 ± 26.0 vs. −2.2 ± 19.4, *p* = 0.095), WOMAC score (9.5 ± 18.1 vs. 0.3 ± 15.7, *p* = 0.091), and HHS (7.0 ± 14.7 vs. −1.0 ± 12.6, *p* = 0.089), at 12 months. Patients with Tonnis 1-2 treated with C-PRP showed a significantly greater improvement in HHS at 12 months of follow-up (Figure 6) compared with those treated with A-PRP (7.0 ± 14.7 vs. 0.3 ± 6.3, *p* = 0.049).

Finally, there were no significant differences in failures and adverse events when comparing the treatments and the OA levels.

## 4. Discussion

The main finding of this study is that C-PRP is a safe approach for the treatment of patients with hip OA, with a low rate of adverse events and failures, although it provided only a mild clinical improvement comparable with A-PRP. In fact, the overall comparative analysis between the two PRPs did not show a significant difference in terms of clinical outcomes, adverse events, and failures at all follow-ups.

The use of intra-articular PRP to address hip OA has the target of modulating the articular environment to reduce inflammation and stimulate anabolism. Platelet alpha-granules contain various growth factors such as cytokines, chemokines, ADP, ATP, calcium ions, histamine, serotonin, dopamine, and up to 800 different proteins [19,20]. In particular, the transforming growth factor beta (TGF-β) preserves and stimulates the proliferation of synoviocytes and the production of hyaluronic acid [21,22], which could have an effect on the progression of OA [10,23,24,25,26]. This may also explain some positive findings on the use of A-PRP, with a significant improvement in particular in the first 3 months. Di Sante et al. [25] and Dallari et al. [10] reported the superiority of PRP treatment over hyaluronic acid in the first months (*p* <0.05) in terms of efficacy. In the latter study there was also a correlation between the clinical course and the OA grade [10]. On the other hand, Battaglia et al. [17] and Doria et al. [26] showed improvement for both groups, but no superiority of PRP over viscosupplementation. Considering the limited and controversial clinical results obtained with A-PRP injections, researchers analyzed new options to better exploit platelet concentrates, investigating homologous blood sources such as the umbilical cord.

The rationale for choosing C-PRP relies on its homologous source, and therefore is not linked to the characteristics of the treated patient. The literature reports how the quantitative and qualitative characteristics of platelets and contained growth factors can be negatively influenced by patient age [27]. PRP from older donors has a more pro-inflammatory composition and can be less active than PRP from younger donors [28,29]. In this light, the use of C-PRP has the theoretical advantage of a greater quantity of growth factors [30] and, from a qualitative point of view, C-PRP was found to contain higher levels of anti-inflammatory molecules than A-PRP, which has instead more pro-inflammatory factors [31]. C-PRP contains molecules of the NK group 2, responsible for the suppression of Natural Killer (NK), Natural Killer T (NKT), and T cells, which is interpreted as a maternal–fetal tolerance mechanism. It also contains a greater amount of growth factors and cytokines, in particular Interleukine-10 (IL-10) which has been correlated with the decrease in VAS pain score [10]. Thanks to these advantages, C-PRP has so far been used successfully in various fields of medicine [32]: in ophthalmology for the treatment of corneal diseases and severe dry eye problems, and in the dermatological field for the healing of wounds and skin lesions. Nevertheless, evidence on the application of C-PRP in orthopedic diseases, including hip OA, is still limited, despite the growing interest in the use of homologous sources of PRP.

Bottegoni et al. [33] examined 60 patients with low and medium knee OA grade and hematopoietic pathologies who were therefore not suitable for injective treatment with A-PRP. For this reason, patients underwent injection with homologous PRP from blood donors, showing a significant improvement in IKDC and KOOS and a significant decrease in VAS at 2 and 6 months. Recently, Caiaffa et al. [34] treated 25 patients suffering from knee OA with a Kellgren–Lawrence grade between 1 and 3 with PRP of cordonal origin, obtaining a significant improvement in clinical scores and a decrease in VAS at 2, 4, and 6 months. However, these studies did not present a comparison group and have a short-term follow-up.

The current comparative study evaluated the safety and efficacy of C-PRP for the treatment of patients with hip OA for the first time, comparing this product with the more commonly used A-PRP. Regarding the primary aim of the study, there were no major adverse events after the injective procedures, which proved to be safe. The only documented complications, including pain and functional limitation in the period immediately following the injection, are common in the injective treatment with PRP [35]. This effect may be attributed to the presence of leukocytes, which could promote an initial inflammatory reaction [36,37,38,39,40]. Although the final overall balance is an anti-inflammatory effect thanks to the inhibition of the NF-kB pathway [41], as also demonstrated in other clinical studies about knee injective therapy [42], leukocytes could cause an initial pain reaction. This, however, is self-limiting and seems not to affect the final treatment outcome [43].

The analysis of the treatment outcome showed an overall modest benefit at short-term follow-up, reaching a statistically significant improvement in the C-PRP group for the VAS pain and the HHS scores for the evaluation of symptoms and function at 2 months. This improvement was not confirmed at 6 and 12 months of follow-up for both scores. In the literature, a progressive worsening of clinical and functional scores has been observed at 6 to 12 months from injection in hip OA [24]. This type of result has been suggested to depend on the individual characteristics of the patient [44]. In particular, it has been emphasized that the best results in PRP treatments can be obtained in young patients, with low BMI and with less cartilage damage [45]. Previous studies have already showed worse results in patients with greater OA levels [10,32]. Similarly, the analysis on influencing factors in this study showed how the OA grade, quantified with the classification of Tonnis, was a determining aspect. In fact, patients with low OA grades (Tonnis 1-2) obtained better clinical results compared with patients with more advanced OA grades (Tonnis 3). Also, when comparing the two treatments only in patient groups with lower OA grades (Tonnis 1-2), C-PRP showed significantly greater improvement in HHS at 12 months of follow-up than A-PRP. Unfortunately, this study presents a heterogeneous population, which limits the possibility to properly investigate subgroups, and these promising results need to be confirmed in future studies.

The relatively small number and the heterogeneity of treated cases are limitations of this study. On the other hand, it was still possible to document the safety of C-PRP treatment and to identify the most responsive subgroups which should be the focus of future studies. The lack of a PRP characterization did not allow the investigation of possible composition differences between the two PRP types and the relation between their compositions and the different clinical outcomes. Future studies should analyze these aspects to optimize the use of different platelet concentrates, as recently underlined also by a consensus of experts suggesting an in-depth coding system for PRP studies [46]. Another limitation is the lack of blinding and randomization. However, all patients underwent an orthobiologics treatment, thus both groups may have been influenced by a similar placebo effect [47,48], and an independent statistician, blinded to the study outcome, selected patients for a match-paired analysis. Finally, patients were enrolled suffering mainly from some functional limitations, but did not present a high symptomatology. Thus, the low level of symptoms such as pain at baseline impaired the possibility to detect large and statistically significant improvements. In this light, these study findings should be interpreted with caution considering the possible biases which could lead to errors in the data interpretation. Nonetheless, C-PRP treatment proved to be safe and short-term benefits were documented despite all these limitations, and future studies should focus on the most promising patient and disease groups that may benefit more from C-PRP for the treatment of hip OA.

## 5. Conclusions

C-PRP injections have shown an excellent safety profile for patients with hip OA, although they offered only a short-term clinical improvement and the comparative analysis did not demonstrate any clinical benefit compared with A-PRP in the general population. However, the results are influenced by the OA grade, with better results in patients with a low grade of OA. When advanced OA cases were excluded, C-PRP showed more benefits than A-PRP. Further studies with larger case series, stronger study design, longer follow-up, and better characterization of patient and disease features that could influence the outcome are needed to confirm the most suitable indications and potential of this biological injective approach.

## Figures and Tables

**Figure 1 jcm-11-04505-f001:**
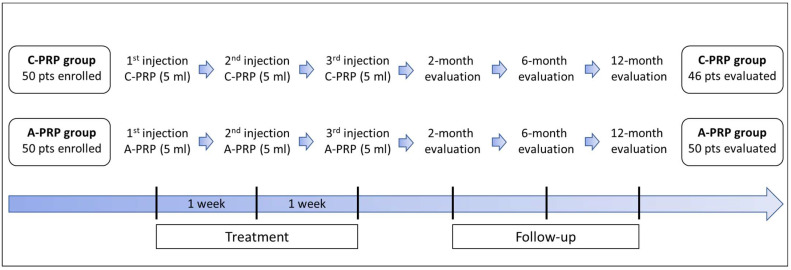
Workflow of the study. A-PRP, autologous platelet-rich plasma; C-PRP, cordonal platelet-rich plasma; pts, patients.

**Figure 2 jcm-11-04505-f002:**
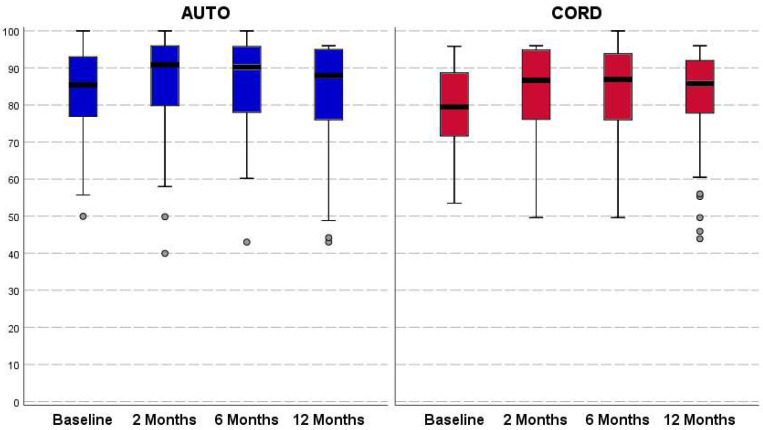
Harris Hip Score (HHS; 0–100 points) trends in both treatment groups. The horizontal black line represents the median, the box limit represents quartiles, and error bars represent 95% Confident Intervals.

**Figure 3 jcm-11-04505-f003:**
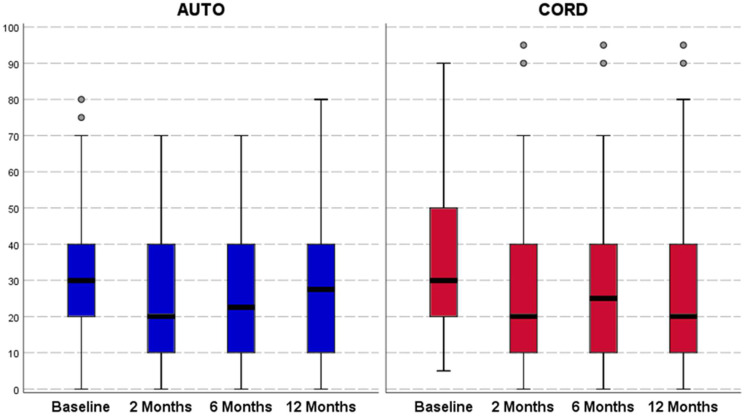
Visual analog scale (VAS; 0–100 points) trends in both treatment groups. The horizontal black line represents the median, the box limit represents quartiles, and error bars represent 95% Confident Intervals.

**Figure 4 jcm-11-04505-f004:**
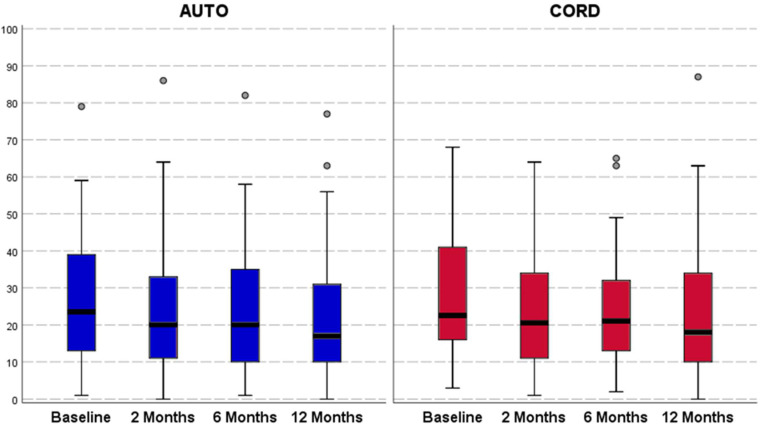
Western Ontario and McMaster Universities Osteoarthritis Index (WOMAC; 0–100 points) trends in both treatment groups. The horizontal black line represents the median, the box limit represents quartiles, and error bars represent 95% Confident Intervals.

**Figure 5 jcm-11-04505-f005:**
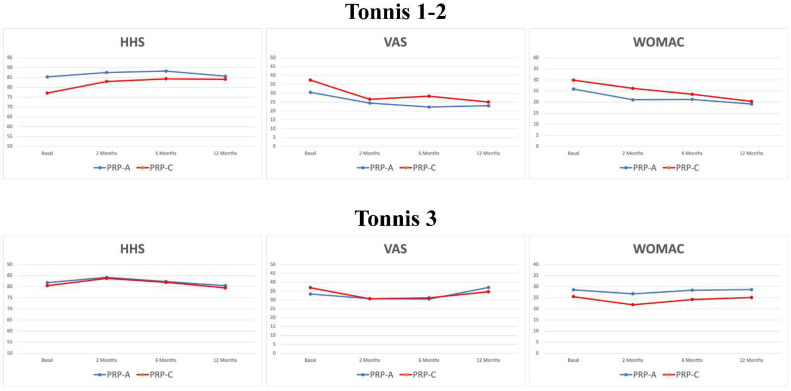
Trends of the evaluated clinical scores based on the OA severity.

**Figure 6 jcm-11-04505-f006:**
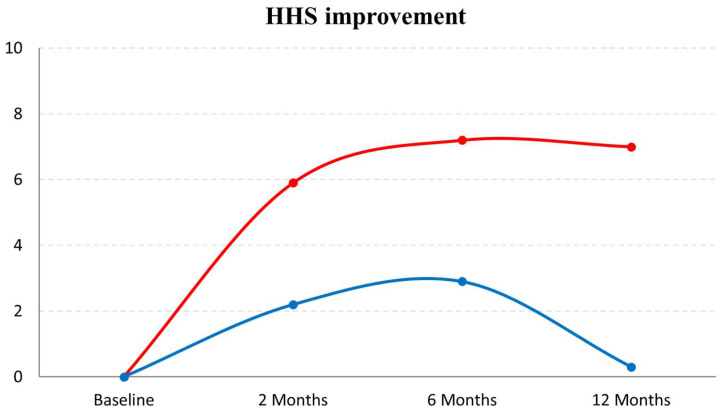
Improvement of the Harris Hip Score (HHS) in patients with Tonnis 1-2. The C-PRP group (red) reported a higher clinical improvement compared with the A-PRP group (blue) at 12 months of follow-up (*p* = 0.049).

**Table 1 jcm-11-04505-t001:** Baseline demographic and clinical characteristics of the included patients of both groups.

Characteristics	C-PRP Group (*n* = 46)	A-PRP Group (*n* = 50)	*p* Value
**Men/Women**	26/20	34/16	n.s.
**Age, y (mean ± SD)**	47.1 ± 11.9	49.5 ± 12.2	n.s.
**Previous Surgery (yes/no)**	16/30	13/37	n.s.
**Tonnis OA Grade**	Grade 1: 6	Grade 1: 5	n.s.
	Grade 2: 19	Grade 2: 23	
	Grade 3: 21	Grade 3: 22	
**VAS pain**	37.2 ± 22.0	31.8 ± 19.9	n.s.
**WOMAC**	27.9 ± 17.2	27.1 ± 17.6	n.s.
**HHS**	78.6 ± 11.4	83.8 ± 11.9	0.026

A-PRP, Autologous PRP; C-PRP, cord PRP; HHS, Harris Hip Score; n.s., not significant; OA, osteoarthritis; SD, standard deviation; VAS, Visual Analogue Scale; WOMAC, Western Ontario and McMaster University Osteoarthritis index; y, years.

**Table 2 jcm-11-04505-t002:** Clinical scores at baseline and follow-ups.

Score	Group	Baseline	2 Months	6 Months	12 Months
**HHS**	** *A-PRP* **	83.8 ± 11.0	86.1 ± 14.1	85.6 ± 12.3	83.4 ± 14.0
	** *C-PRP* **	78.6 ± 11.4	83.3 ± 13.3	83.2 ± 13.1	82.0 ± 14.2
**VAS pain**	** *A-PRP* **	31.8 ± 19.9	27.2 ± 18.8	25.9 ± 18.0	29.2 ± 21.0
	** *C-PRP* **	37.17 ± 22.0	28.5 ± 22.1	29.6 ± 22.1	29.4 ± 24.6
**WOMAC**	** *A-PRP* **	27.1 ± 17.6	23.6 ± 17.4	24.4 ± 17.7	23.3 ± 18.2
	** *C-PRP* **	27.9 ± 17.2	24.2 ± 17.2	23.9 ± 15.8	22.5 ± 18.3

A-PRP, Autologous PRP; C-PRP, cord PRP; HHS, Harris Hip Score; VAS, Visual Analogue Scale; WOMAC, Western Ontario and McMaster University Osteoarthritis index. There was no statistically significant difference between the two groups in terms of improvement and absolute value, except for the baseline HHS which was higher in the C-PRP group (*p* = 0.026).

## Data Availability

Not applicable.

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
