# Peer review of "Umbilical Cord PRP vs. Autologous PRP for the Treatment of Hip Osteoarthritis"

_jcm, 2022, doi:10.3390/jcm11154505_

Round 1

Reviewer 1 Report

Dear Authors!

The manuscript is well written and I have no any major concerns about it.

The manuscript is devoted to contemporary problem - finding the new possible treatment approach for hip osteoarthritis.

There no known approved medications for hip osteoarthritis and the only one effective treatment is total hip arthroplasty.

Authors provided the new treatment approach based on cells technologies - and compare two treatment arms with cPPR and A-PPR.

The c-PPR looks like more promising option for patients with hip osteoarthritis.

The manuscript has a proper design, statistics and outcomes assessment, the quality of English is appropriate.

Author Response

Dear reviewer,

Thank you for your positive revision.

We sent the manuscript for the correction of the English language to a professional English translator for the editing service.

We hope the corrected manuscript is now more interesting for the readers and suitable for publication in the Journal of Clinical Medicine. 

Reviewer 2 Report

1.      Keywords need to reorder based on alphabetical order.

2.      Other than abbreviations, each keyword must use lowercase letters.

3.      The present manuscript is lack novelty and scientific contribution. In the literature, Umbilical Cord PRP and Autologous PRP have been widely studied in the past. The authors should explain their novelty more advance in the introduction section.

4.      Research related to hip osteoarthritis should be explained with their findings and its limitation to show the present research gap that was filled in trough present study.

5.      Osteoarthritis at a certain level brings pain to the patient that needs a total hip implant as a solution to solve the problems. The authors need to deliver these important points. Also to support this explanation, additional references published by MDPI is should be adopted as follows: Computational Contact Pressure Prediction of CoCrMo, SS 316L, and Ti6Al4V Femoral Head against UHMWPE Acetabular Cup under Gait Cycle. J. Funct. Biomater. 2022, 13, 64. https://doi.org/10.3390/jfb13020064

6.      In the materials and methods section, the author should include the research workflow in the materials and methods section for better understanding.

7.      Tools information included to its manufacturer should be given.

8.      Error and tolerance of the present study should be included.

9.      The present study uses a very small number of involved subjects which would bring biased results. The authors need to increase their sample since the involved sample impacting to the results.

10.   The English used needs proofreading due to grammatical issues and language style.

11.   The present conclusion is not solid, further elaboration is needed.

12.   Further study needs to be explained in the conclusion section.

Author Response

Dear reviewer, 

Thank you for allowing us to improve our manuscript. We modified our manuscript following your suggestions and advice. Please see the following point by point answers to your comments and the related changes.

  1. Keywords need to reorder based on alphabetical order.

- Done (lines 30-31).

  1. Other than abbreviations, each keyword must use lowercase letters.

- Done (lines 30-31).

  1. The present manuscript is lack novelty and scientific contribution. In the literature, Umbilical Cord PRP and Autologous PRP have been widely studied in the past. The authors should explain their novelty more advance in the introduction section.

- We agree with reviewer that autologous PRP has been widely studied over the years. Our groups already published one of the most important RCTs on the use of autologous PRP for the treatment of hip OA. On the other side, umbilical cord PRP has been used so far successfully in various fields of medicine, ranging from ophthalmology to dermatology. However, evidence on the application of C-PRP in orthopedic diseases, including hip OA, is still limited. We specified this in the introduction section (lines 63-65).

Therefore, this comparative study represents the first study in the literature that evaluated the safety and efficacy of C-PRP for the treatment of patients with hip OA, comparing this product with the more commonly used A-PRP. For this reason, we consider this study of important clinical relevance, as it is the only available one on this topic and provides important information on the use of this innovative product in patients with hip OA (lines 324-326).

  1. Research related to hip osteoarthritis should be explained with their findings and its limitation to show the present research gap that was filled in trough present study.

- Research on conservative injectable options for hip OA already showed some findings of different treatments, although with sometime conflicting or limited results. For this reason, there are research efforts toward new treatment options. This is the case of c-PRP, and this study fills the gap of the literature in documenting potential and limitations of c-PRP use for hip OA (lines 40-44 and 47-48).

  1. Osteoarthritis at a certain level brings pain to the patient that needs a total hip implant as a solution to solve the problems. The authors need to deliver these important points. Also, to support this explanation, additional references published by MDPI is should be adopted as follows: Computational Contact Pressure Prediction of CoCrMo, SS 316L, and Ti6Al4V Femoral Head against UHMWPE Acetabular Cup under Gait Cycle. J. Funct. Biomater. 2022, 13, 64. https://doi.org/10.3390/jfb13020064

- Thank you for your comment. We address these point in the introduction section (lines 42-45). Also, we added this reference as requested by the reviewer (line 45).

  1. In the materials and methods section, the author should include the research workflow in the materials and methods section for better understanding.

- Done (Figure 1).

  1. Tools information included to its manufacturer should be given.

- Both C-PRP and A-PRP were produced with a manual method of the hematology department and without the use of commercial kits. We specified this aspect in the method section (lines 120-121 and 135).

  1. Error and tolerance of the present study should be included.

- We increased the limitation section of this study to underline the possible errors of this analysis and how the related findings should be interpreted with caution (lines 362-364).

  1. The present study uses a very small number of involved subjects which would bring biased results. The authors need to increase their sample since the involved sample impacting to the results.

- We agree with the reviewer that the small number of patients is a limitation of our study. Nevertheless, analyzing the available literature on the injective treatment of hip OA with orthobiologics like PRP, the number of patients included in this study is relatively high, considering that only two studies exceeded 100 patients. Thus, while larger studies are welcomed, this field will benefit significantly from these early data, as they start filling a gap in the literature of an otherwise undocumented aspect. Anyway, we discussed this aspect in the limitation section, as correctly underlined by the reviewer (lines 353-354).

  1. The English used needs proofreading due to grammatical issues and language style.

- We sent the manuscript for the correction of the English language to a professional English translator for the editing service.

  1. The present conclusion is not solid, further elaboration is needed.

- Done (lines 370-378).

  1. Further study needs to be explained in the conclusion section.

- Done (lines 370-378).

Round 2

Reviewer 2 Report

I recommend this manuscript for acceptance.

Author Response

Dear reviewer,

Thank you for your positive revision.

We think that this study is now suitable for the publication in the Journal of Clinical Medicine.